# Glycoprotein E-Displaying Nanoparticles Induce Robust Neutralizing Antibodies and T-Cell Response against Varicella Zoster Virus

**DOI:** 10.3390/ijms25189872

**Published:** 2024-09-12

**Authors:** Hong Wang, Sibo Zhang, Wenhui Xue, Yarong Zeng, Liqin Liu, Lingyan Cui, Hongjing Liu, Yuyun Zhang, Lin Chen, Meifeng Nie, Rongwei Zhang, Zhenqin Chen, Congming Hong, Qingbing Zheng, Tong Cheng, Ying Gu, Tingting Li, Ningshao Xia, Shaowei Li

**Affiliations:** 1State Key Laboratory of Vaccines for Infectious Diseases, Xiang An Biomedicine Laboratory, School of Public Health, School of Life Sciences, Xiamen University, Xiamen 361102, China; 2State Key Laboratory of Molecular Vaccinology and Molecular Diagnostics, National Institute of Diagnostics and Vaccine Development in Infectious Diseases, Collaborative Innovation Center of Biologic Products, National Innovation Platform for Industry-Education Integration in Vaccine Research, The Research Unit of Frontier Technology of Structural Vaccinology of Chinese Academy of Medical Sciences, Xiamen University, Xiamen 361102, China

**Keywords:** nanoparticle, Varicella zoster virus, glycoprotein E, neutralizing antibody, T-cell response

## Abstract

The Varicella zoster virus (VZV), responsible for both varicella (chickenpox) and herpes zoster (shingles), presents significant global health challenges. While primary VZV infection primarily affects children, leading to chickenpox, reactivation in later life can result in herpes zoster and associated post-herpetic neuralgia, among other complications. Vaccination remains the most effective strategy for VZV prevention, with current vaccines largely based on the attenuated vOka strains. Although these vaccines are generally effective, they can induce varicella-like rashes and have sparked concerns regarding cell virulence. As a safer alternative, subunit vaccines circumvent these issues. In this study, we developed a nanoparticle-based vaccine displaying the glycoprotein E (gE) on ferritin particles using the SpyCatcher/SpyTag system, termed FR-gE. This FR-gE nanoparticle antigen elicited substantial gE-specific binding and VZV-neutralizing antibody responses in BALB/c and C57BL/6 mice—responses that were up to 3.2-fold greater than those elicited by the subunit gE while formulated with FH002C, aluminum hydroxide, or a liposome-based XUA01 adjuvant. Antibody subclass analysis revealed that FR-gE produced comparable levels of IgG1 and significantly higher levels of IgG2a compared to subunit gE, indicating a Th1-biased immune response. Notably, XUA01-adjuvanted FR-gE induced a significant increase in neutralizing antibody response compared to the live attenuated varicella vaccine and recombinant vaccine, Shingrix. Furthermore, ELISPOT assays demonstrated that immunization with FR-gE/XUA01 generated IFN-γ and IL-2 levels comparable to those induced by Shingrix. These findings underscore the potential of FR-gE as a promising immunogen for the development of varicella and herpes zoster vaccines.

## 1. Introduction

Varicella zoster virus (VZV) is a human alphaherpesvirus and causes two distinct diseases: varicella (also known as chickenpox) and herpes zoster (HZ, shingles) [1]. Primary infection with VZV can cause varicella. After an incubation period of 10–21 days, a widely distributed chickenpox rash appears, which is highly contagious [2]. Varicella occurs most commonly in infants and children, and the symptoms are more severe in adults [3]. According to conservative estimates by the World Health Organization (WHO), there are approximately 4.2 million cases of varicella-related severe complications worldwide each year, resulting in hospitalization [4]. Secondary infection with VZV can lead to herpes zoster (HZ), which is usually characterized by a painful banded rash on and postherpetic neuralgia (PHN), accompanied by many complications including keratitis and encephalitis and mostly occurs in middle-aged and older people over 50 years old [5,6]. A study of 21 counties in Europe indicates that the incidence of HZ varies from 2‰ to 4.6‰ per year [7].

Vaccination is considered the most effective and cost-efficient method to prevent VZV infection and its potential complications [8]. Both varicella and herpes zoster can be prevented by live attenuated virus vaccines [9]. As early as 1971, the Japanese scholar Michiaki Takahashi isolated the VZV Oka strain from a varicella patient, and developed the Oka strain attenuated varicella vaccine in 1974 [10]. Most chickenpox vaccines have taken the technical route of live attenuated vaccines based on the vOka strain. As the dose of the varicella vaccine increases, the effectiveness of the vaccine is enhanced, and the severity of the rash from breakthrough infections gradually decreases [11,12,13]. Overall, individuals with breakthrough infections typically exhibit milder symptoms than those who are unvaccinated [13]. Cell-mediated immunity (CMI) is proven to be necessary and effective for the prevention of VZV reactivation and the development of HZ [14,15]. Zostavax, the first live attenuated vaccine against HZ and PHN, was shown to reduce the incidence of shingles by 51.3% and PHN by 66.5% [16]. However, the vOka strain has the ability to establish latency in human nerve cells and reactivate, leading to herpes zoster, which has raised wide safety concerns [17,18,19,20,21]. Additionally, the skin tropism of the vOka strain may result in the occurrence of varicella-like rashes after vaccination [18,22,23]. Subunit vaccines offer an excellent way to solve this problem. The VZV membrane glycoprotein gE plays a crucial role in viral replication and assembly, and it is an indispensable glycoprotein in the process of VZV T-cell infection. This also makes it a popular target for VZV vaccine development. Shingrix, a commercially available herpes zoster vaccine approved by the FDA in 2017, utilizes gE as the antigen in combination with the AS01B adjuvant [24]. This vaccine has demonstrated a high efficacy of 97.2% in protecting against herpes zoster and an impressive 91.2% in preventing post-herpetic neuralgia in a population aged 50 and above [25].

Nanoparticles are a promising vaccine form for use against various pathogens. Nanoparticle vaccines can improve antigen stability, facilitate targeted delivery to antigen-presenting cells (APCs), and offer a higher level of safety compared to traditional vaccines [26]. Ferritin is widely used in nanoparticle vaccine design. Several nanoparticle vaccines are currently in development, including those based on the artificially designed I53 particle and an influenza vaccine employing hemagglutinin-based nanoparticles in clinical trials. Additionally, vaccines for the treatment of respiratory syncytial virus (RSV) and SARS-CoV-2, utilizing polymerized nanoparticles, are in preclinical stages [27,28,29,30]. Other antigen display strategies, such as chemical cross-linking, genetic fusion, and tag coupling [31,32,33], are used to display antigen protein on the surface of particle carriers. The SpyTag/SpyCatcher split protein system has been developed to conjugate antigens onto nanoparticles covalently [31,34]. The nanoparticle platform based on the SpyTag/SpyCatcher system is capable of providing high-density, unidirectional antigen display [35]. Some research shows that the Spy system combined with a nanoparticle platform can increase binding antibody titers against West Nile virus (4-fold) and neutralizing titers against SARS-CoV-2 (10- to 120-fold) [36,37]. SpyTag/SpyCatcher can be genetically fused to different antigens and expressed across various expression systems [31,38,39]. Additionally, the split proteins are small and unlikely to affect the folding of the antigen, and SpyCatcher has been demonstrated to enhance the solubility of typically recalcitrant antigens [31].

To overcome the above limitation of the varicella vaccine and provide alternative vaccine candidates, we designed and evaluated a VZV nanoparticle vaccine that covalently conjugates gE subunits to self-assembled ferritin. This approach facilitated the display of gE on the surface of ferritin particles. The newly constructed particulate gE antigen demonstrated the ability to stimulate a significantly stronger neutralizing antibody response and T-cell response in mice. Our results will benefit the antigen design of next-generation vaccines for varicella and HZ.

## 2. Results

### 2.1. Construction and Purification of the FR-gE Nanoparticle

To augment the immunogenicity of gE-based vaccines, we engineered a gE-SpyTag (gE-ST) protein for presentation on a ferritin nanoparticle. The gE ectodomain (ECD, residues 23-537) was modified at the N-terminus with a melittin signal peptide (MSP) for enhanced secretion and a hexahistidine (6×His) tag to facilitate affinity purification. Additionally, a Spytag (ST, 13 amino acids) derived from *Streptococcus pyogenes* was appended to the C-terminus of the gE ECD (Figure 1A). Concurrently, SpyCatcher (SC, 138 amino acids), also from *Streptococcus pyogenes*, was fused to the N-terminus of *Helicobacter pylori* ferritin through a flexible 3×GGGGS linker to minimize steric hindrance affecting particle assembly, resulting in the construct Ferritin–SpyCatcher (FR-SC) (Figure 1A). The proposed assembly of FR-SC with gE-ST is illustrated in Figure 1B, with each FR-SC subunit capable of binding to a gE-ST protein, thus forming the nanoparticulate gE antigen, hereafter referred to as FR-gE.

The gE-ST gene was inserted into the pIEX plasmid vector downstream of the p10 promoter, while FR-SC was cloned into the pTYB11 vector. gE-ST was expressed in High Five insect cells using a baculovirus system to ensure proper protein folding and later purified from the culture medium. Conversely, FR-SC was produced in *E. coli* and purified through thermal denaturation and ammonium sulfate precipitation. Following centrifugation, the supernatants were processed via metal affinity chromatography using a Ni-NTA resin. On SDS-PAGE, the gE-ST proteins exhibited a molecular weight of approximately 70 kDa, verified by the gE-specific monoclonal antibody 1B11-HRP (Figure 1C). FR-SC proteins, extracted solubly from *E. coli*, showed a molecular weight of 40 kDa after purification (Figure 1E). Purified gE-ST and FR-SC were mixed at 37 °C for 1 h at a 3:1 molar ratio and further purified with Superdex 200 increase chromatography to isolate FR-gE (Figure 1D). Fractions collected with retention volumes of 9.2 mL underwent SDS-PAGE, revealing that FR-gE, isolated as the peak 1 fraction in size-exclusion chromatography (SEC), was highly pure and devoid of significant contaminant proteins (Figure 1E).

### 2.2. Physiochemical Properties, Structure Analysis, and Antigenicity Evaluation of FR-gE

We subsequently assessed the physicochemical properties of FR-gE in comparison with gE-ST and FR-SC. Sedimentation velocity analytical ultracentrifugation (SV-AUC) was employed to determine the sedimentation coefficients of these proteins (Figure 2A). The c(S) profile revealed a prominent sedimentation coefficient for FR-gE of 33.2 S, while the coefficients for FR-SC and gE-ST were 19.8 S and 3.74 S, respectively, indicating a substantial increase in the molecular mass of FR-gE. Additionally, dynamic light scattering (DLS) was conducted to ascertain the diameters of the FR-SC and FR-gE particles. FR-SC particles had an average diameter of approximately 10 nm (Figure 2B), which increased to nearly 30 nm for FR-gE upon binding with gE-ST (Figure 2C), suggesting that gE-ST effectively binds to FR-SC, enlarging the particle size.

The morphology of the particles was further analyzed using transmission electron microscopy (TEM) with negative staining. FR-SC demonstrated a clear and homogenous morphology (Figure 2D), whereas FR-gE particles were encircled by a subtle, blurred halo indicative of the gE-ST protein (Figure 2E). To decode the structure of FR-gE, cryo-electron microscopy (cryo-EM) analysis was conducted on both FR-gE and FR-SC. Two-dimensional (2D) classification of the raw micrographs illustrated the gE-ST protein on the surface of FR-SC particles (Figure 2F). Advancing from this, we generated a three-dimensional (3D) reconstruction map from these images and created a 3D model of FR-gE (Figure 2G). This reconstruction revealed pronounced surface density corresponding to the gE protein, confirming the successful display of gE-ST on the ferritin nanoparticle.

In a further study, the antigenic properties of FR-gE and gE were probed using ELISA with six mouse monoclonal antibodies (mAbs) specific to gE. The results revealed that FR-gE and gE both show strong reactivities to all the tested mAbs, with the half-maximal effective concentrations (EC_50_) ranging from 10.1 to 29.5 ng/mL and 5.0 to 36.6, respectively, ascertained at the nanogram level with no apparent change (Figure 2H). These physicochemical and bioactivity characterization of FR-gE revealed an increased molecular mass and particle diameter compared to gE-ST and FR-SC, with successful gE-ST display on the ferritin nanoparticle surface confirmed through cryo-EM analysis and strong antigenicity demonstrated by ELISA.

### 2.3. FR-gE Induces Th1-Skewed and High Neutralizing Antibody Response in BALB/c Mice

To assess the immunogenicity of FR-gE, groups of BALB/c mice (*n* = 5 per group) were immunized with FR-gE or gE-ST at weeks 0 and 2, combined with FH002C—a nitrogen bisphosphonate-modified zinc-aluminum hybrid adjuvant [40]. The dosage for gE was 0.5 µg per administration, whereas FR-gE was administered at dosages of 0.5 µg and 5 µg of gE proteins (0.78 µg and 7.8 µg of FR-gE) to evaluate a potential dose–response effect. Antigen-specific IgG and VZV-specific neutralizing antibody titers in mouse sera at weeks 1 through 4 were determined using endpoint ELISA and VZV neutralization assays, respectively. The 0.5 µg FR-gE and gE groups showed seroconversion at week 2 post-vaccination, in contrast to the 5 µg FR-gE group, where seroconversion occurred by week 1. gE-specific IgG titers for the FR-gE groups were higher than those for gE at weeks 2 and 3 (approximately 2.5- and 2.8-fold, respectively) but slightly lower at week 4, with no significant differences. The 5 µg FR-gE group maintained consistently higher gE-specific IgG titers than the 0.5 µg group, by 3.6- to 5.85-fold (Figure 3A).

To categorize IgG antibody subclasses, we measured IgG1 (Figure 3B) and IgG2a (Figure 3C) titers in week 4 sera. IgG1 titers did not differ significantly across groups, but the 5 µg FR-gE group exhibited elevated IgG2a titers relative to the other groups, with gE alone not raising IgG2a titers above the detection threshold (1000). Additionally, the ratios of IgG1 to IgG2a suggested more pronounced T helper (Th) 1-biased responses in two FR-gE groups (Figure 3D). For comparable dosages, FR-gE elicited similar total IgG and IgG1 levels to gE yet induced stronger IgG2a responses, indicating a potentially greater Th1-skewed immune activation by FR-gE compared to gE alone.

Neutralizing antibody titers are important for the varicella vaccine. We thus measured neutralizing antibody titers using a neutralizing assay at week 4. The result showed that, at equivalent gE doses (0.5 µg), mice immunized with FR-gE produced neutralizing antibody titers higher than those of the subunit gE-ST group by ~2-fold (Figure 3E). These findings demonstrated that the particulate antigen FR-gE, when paired with the adjuvant FH002C, prompts a superior neutralizing antibody response compared to the subunit antigen gE-ST.

### 2.4. FR-gE Induces Robust Humoral and Cellular Response in C57BL/6 Mice

To further evaluate particulate gE as an antigen for both varicella and shingles vaccines, we designed a combination of two well-characterized adjuvants, the traditional aluminum adjuvant for varicella and the potent XUA01 adjuvant (a mimic of licensed liposome-based AS01B adjuvant) for shingles, and compared it with the varicella attenuated vaccine, shingles attenuated vaccine, and recombinant shingles vaccine. For further detection of cellular immune responses, we performed an immunization assay in C57BL/6 mice (*n* = 5 per group), administering gE-ST and FR-gE in combination with an XUA01 adjuvant as well as an aluminum hydroxide adjuvant at weeks 0 and 4. As controls, we used a freeze-dried varicella live attenuated vaccine and licensed Shingrix vaccine, with an immunization dose of 5 μg of gE protein per dose for both the test groups and Shingrix. The human dose for the freeze-dried shingles vaccine was 19,400 PFU, so we administered a scaled mouse dose of approximately 200 PFU for the varicella vaccine and 2000 PFU for the HZ vaccine. We measured gE-specific IgG titers in mouse sera at weeks 2, 4, and 6 via endpoint ELISA (Figure 4A), observing a decrease in gE-specific binding antibody titers at week 4, which suggested that the peak of primary immune response occurred between weeks 2 and 4. Post-booster immunization at week 4 resulted in peak antibody titers at week 6 ranging from 6-log to 7-log. Throughout the experiment, both gE-ST and FR-gE paired with the XUA01 adjuvant elicited stronger immune responses than those paired with the aluminum hydroxide adjuvant, with no significant difference. At week 6, the average titers in the 2000 PFU and 200 PFU groups remained at a lower level of 4–5 log, significantly lower than those of FR-gE with the aluminum hydroxide adjuvant or XUA01. Notably, the gE-specific IgG titers induced by FR-gE with the XUA01 adjuvant were up to 6.5-log, comparable to the Shingrix group, with no significant difference.

In the sixth week, we assessed the titers of the IgG1 (Figure 4B) and IgG2a (Figure 4C) subclasses. Mice immunized with FR-gE/XUA01 and Shingrix produced higher IgG1 and IgG2a titers, with no significant difference when compared to groups with aluminum hydroxide adjuvant; however, the XUA01 adjuvant groups exhibited notably higher IgG2a titers, particularly with FR-gE. We calculated the ratio of IgG1 to IgG2a titers for each mouse (Figure 4D) and found low titers in the 2000 and 200 PFU groups, implying limited significance for the IgG1/IgG2a ratios. Conversely, the FR-gE/XUA01 and Shingrix groups showed lower IgG1/IgG2a ratios, indicative of a stronger Th1-biased immune response. The other three groups (gE-ST/XUA01, gE-ST/Al, FR-gE/Al) had higher ratios, suggesting a Th2-biased immune response.

Next, we measured the neutralizing antibody titers of the sixth-week sera using a VZV neutralization assay (Figure 4E). Mice immunized with FR-gE/XUA01 displayed the highest neutralizing antibody titers, significantly surpassing the gE-ST/Al and FR-gE/Al groups, with non-significant increases of approximately 3.2-fold compared to the gE-ST/XUA01 group and 2.1-fold relative to Shingrix. Both FR-gE groups generated higher neutralizing antibody titers than subunit gE-ST groups with two different adjuvants (Al and XUA01), suggesting particulate antigen FR-gE can further enhance the neutralizing titer produced in mice.

Considering the importance of the CMI response for herpes zoster vaccine development, we used ELISPOT to measure levels of interferon-gamma (IFN-γ) (Figure 4F) and interleukin-2 (IL-2) (Figure 4G) after the booster, comparing only the FR-gE/XUA01, 2000 PFU, Shingrix, and saline groups based on the preceding outcomes of IgG1/IgG2a ratios (Figure 4D). The results indicated that FR-gE with XUA01 generated high amounts of IFN-γ and IL-2, comparable to those induced by Shingrix and significantly higher than the other two live attenuated vaccines, suggesting that FR-gE holds potential as an immunogen candidate for a herpes zoster vaccine.

## 3. Discussion

At least 36 countries and regions worldwide have included varicella vaccines in their immunization programs, but China has not yet included them in its national immunization program [41]. Currently, the development of a varicella vaccine is mainly based on the technical route of attenuated live vaccines. The skin tropism of the vOka strain also results in the frequent occurrence of a varicella-like rash as a common adverse event after vaccination [18,22,23]. Considering these potential risks, the administration of the vOka varicella vaccine is typically contraindicated in immunocompromised individuals [18,42]. Therefore, a safer vaccine would be beneficial in establishing population immunity against varicella throughout the world. Previously, a skin- and neuro-attenuated live vaccine v7D has completed preclinical studies [43]. Although v7D is expected to have clinical advantages over vaccines based on the vOka strain, such as a lower incidence of varicella-like rash and vaccine strain-induced cases of herpes zoster, recombinant protein vaccines or nanoparticle-based vaccines can perfectly address these concerns. Additionally, recombinant protein vaccines are favored for their safety as they do not contain live organisms and are capable of inducing a protective immune response against specific pathogens, while nanoparticle-based vaccines can further enhance antigen stability, ensure targeted delivery, and boost immunogenicity, which broaden their spectrums of application prospects.

In this study, we used the baculovirus expression vector system (BEVS) to express the VZV glycoprotein E, and constructed a novel particle-based vaccine antigen called FR-gE, which can stimulate strong neutralizing antibody titers in mice, and unexpectedly demonstrated cellular immune responses. Compared with other expression systems such as *Escherichia coli*, yeast, and mammalian cells, the BEVS has the advantage of safety, as post-translational modifications (such as glycosylation, phosphorylation, acetylation, etc.) can be performed on recombinant proteins for proper protein folding [44]. Currently, the market features eight human vaccines such as Flublok^®^ for influenza virus, Cervarix™ for papillomavirus, NVX-CoV2373 for SARS-CoV-2, and five veterinary vaccines developed using the BEVS, with several more in clinical development, highlighting the BEVS’s efficacy in meeting research and industrial needs for vaccine antigen production [45].

To date, there are no reports of available recombinant nanoparticle vaccines for varicella or shingles. In this research, we constructed a particulate antigen FR-gE. Regardless of which adjuvant (FH002C, aluminum hydroxide adjuvant, or AS01B-like adjuvant XUA01) was used, mice immunized with FR-gE produced higher levels of gE-specific binding antibody titers, indicating that the particle-based antigen FR-gE possessed stronger antigenicity than subunit gE proteins. Neutralizing antibodies, considered an important indicator of varicella vaccine effectiveness, were raised to approximately 2.0, 2.6, and 3.2 times higher than gE-ST with FH002C, aluminum hydroxide, and XUA01 adjuvants, respectively, and 2.1 times higher than Shingrix by FR-gE. The more intuitive comparison of the efficacies of different vaccine combinations are shown in Table 1. These results showed that the particle-based antigen FR-gE could stimulate higher levels of neutralizing antibodies compared to subunit antigens gE-ST, indicating that FR-gE is more suitable as an antigen candidate for a novel varicella vaccine.

Interestingly, the results of antibody subclass analysis showed a lower IgG1/IgG2a ratio induced by FR-gE comparable to that induced by gE-ST when both FH002C and XUA01 adjuvants were used. A previous study showed that IgG1 and IgG2a are markers of Th2- and Th1-biased responses, respectively [46]. The IgG1/IgG2a ratio is also believed to be related to the bias towards Th1 or Th2 immune responses [47]. Therefore, the lower IgG1/IgG2a ratio exhibited by FR-gE suggests its potential to activate Th1 immune responses. This hypothesis was further supported by observing that mouse splenocytes immunized with FR-gE/XUA01 and Shingrix exhibited equivalent cytokine levels. Some studies have reported that VLP epitopes can be presented to dendritic cells via MHC I and II, leading to the activation of both helper and cytotoxic T cells [48]. These findings also suggest that the particle-based antigen FR-gE not only activates strong humoral immune responses but also possesses the potential to activate cellular immune responses. However, whether FR-gE can elicit antigen-specific cytotoxic T-lymphocyte responses remains to be verified.

In summary, we have developed a particulate antigen, FR-gE, based on the VZV glycoprotein E, which can induce stronger immunogenicity in mice compared to subunit antigens and the attenuated varicella vaccine, as well as a cellular immune response comparable to that of Shingrix. This also opens up broader avenues for the research and development of vaccines against varicella and herpes zoster.

## 4. Materials and Methods

### 4.1. Cell Lines and Virus Strains

Spodoptera frugiperda (Sf9) insect cells (Thermo Fisher Scientific, Waltham, MA, USA) and High Five insect cells (Thermo Fisher Scientific, USA) were used for transfection and protein expression. Human acute retinal pigment epithelial cells (ARPE-19, ATCC, Manassas, VA, USA) and VZV vOka strains (Beijing WanTai BioPharm, Beijing, China) were used in the VZV-neutralizing assay.

### 4.2. Protein Expression and Purification

The gE-ST plasmids were co-transfected into Sf9 insect cells with v-cath/chiA gene-deficient baculovirus DNA for the generation and amplification of recombinant baculoviruses, which then infected Hive Five insect cells to express the recombinant proteins. The gE-ST proteins were solubly expressed and secreted into the culture medium. The centrifugation supernatants of the cell cultures were subjected to metal affinity chromatography using Ni Sepharose 6 Fast Flow (Cytiva, Marlborough, MA, USA) resin with 50 mM and 250 mM imidazole to elute contaminant proteins and gE-ST, respectively. The FR-SC proteins were transferred into ER2566 Competent Cells (Weidi Bio, Shanghai, China) to express the recombinant protein and purified by thermal denaturation at 70 °C for 10 min and ammonium sulfate precipitation. The purified gE-ST and FR-SC were co-incubated at 37 °C for 1 h at a molar ratio of 3:1, allowing Spycatcher/Spytag pairing by chemical conjugation, and the mixture was then purified by Superdex 200 increase to isolate FR-gE.

### 4.3. SDS-PAGE and Western Blot

Protein samples mixed with 6× loading buffer (300 mM Tris·HCl pH6.8, 12% SDS, 0.6% bromophenol blue, 600 mM DTT, 12% SDS, and 60% glycerol) were boiled for 10 min, loaded on 12% gradient SDS-polyacrylamide gel, and electrophoresed for 30 min at 180 V. The gel was stained with Coomassie Brilliant Blue R-250 (Bio-Rad, Shanghai, China) for 20 min at room temperature. For Western blots, gel was transferred to a nitrocellulose membrane (Whatman, Dassel, Germany) using a Trans-Blot Turbo transfer system (Bio-Rad, China). The membrane was blocked and incubated with the gE HRP-conjugated monoclonal antibody 1B11-HRP (1:5000 dilution). Excess antibodies were removed by five 5 min washes. The membrane was then developed by using SuperSignal ELISA Pico Chemiluminescent Substrate Kit (Thermo Fisher Scientific, USA).

### 4.4. Size-Exclusive Chromatography (SEC)

Purified gE-ST and FR-SC were mixed in a 3:1 ratio and incubated for 1 h at 37 °C to prepare FR-gE protein complexes. Protein complexes were further purified using Superdex 200 increase (Cytiva, USA) by the AKTA system (Cytiva, USA) at a flow rate of 0.7 mL/min. The fractions were harvested and analyzed by SDS-PAGE.

### 4.5. Analytical Ultracentrifugation (AUC)

The AUC assay was performed using a Beckman XL-Analytical ultracentrifuge (Beckman Coulter, Brea, CA, USA). The sedimentation velocity (SV) was carried out at 20 °C. The AN-60 Ti rotor speed was set to 30,000 rpm according to the molecular weight of the proteins, and proteins were diluted to about 1 mg/mL with PBS. Data were collected using SEDFIT 16.1c computer software. Multiple curves were fit to calculate the sedimentation coefficient (S) using the continuous sedimentation coefficient distribution model [c(s)]. The c(s) was then used to estimate protein molar mass.

### 4.6. Dynamic Light Scattering (DLS)

The cuvette specially used for dynamic light scattering experiments was repeatedly rinsed more than 20 times with ultra-pure water. Proteins were diluted to 0.1–0.5 mg/mL and centrifuged at 13,000 rpm for 3 min. Protein samples were pipetted into a cuvette and measured by a highly sensitive nanoparticle analyzer (Brookhaven, Nashua, NH, USA). The incident laser wavelength was set as 824 nm, and the Regulation algorithm was used for calculations.

### 4.7. Electron Microscopy

The proteins FR-SC and FR-gE were analyzed by transmission electron microscopy (TEM). Briefly, samples were absorbed onto 200-mesh carbon-coated copper grids for 1 min. Then, the grids were rinsed with phosphotungstic acid for 45 s and blotted with absorbent paper. Specimens were evaluated by JEM-2100HC (EOL, Osaka, Japan).

### 4.8. Cryo-EM Sample Preparation and Data Collection

For the cryo-EM grid preparation, 3 μL of purified FR-gE at a concentration of 2.30 mg/mL was applied to glow-discharged (80 s at 20 mA) holey carbon Quantifoil grids (R1.2/1.3, 300 mesh, Quantifoil Micro Tools, Jena, Germany). Subsequently, the grids were plunge-frozen using a Vitrobot Mark IV (Thermo Fisher Scientific, USA) under conditions of 100% humidity and a temperature of 4 °C. The datasets were acquired on a Tecnai F30 transmission electron microscope (Thermo Fisher Scientific, USA) operating at 300 kV and equipped with a Gatan K3 direct electron detector. Image collection was facilitated using the SerialEM version 3.8.5 software at a nominal magnification of 39,000× in super-resolution mode, with a pixel size of 0.389 Å and an accumulated electron dose of 60 e^−^·Å^−2^.

### 4.9. Image Processing and 3D Reconstruction

Image processing and 3D reconstruction were performed within cryoSPARC v3 [49]. All dose-fractioned images were motion-corrected and dose-weighted by MotionCor2, with the estimation of their contrast transfer function (CTF) parameters performed using Patch CTF estimation [50]. Particle selection involved the use of both the “Blob picker” and “Template picker”, followed by multiple rounds of reference-free 2D classifications. The selected good particles underwent ab initio reconstruction and subsequent non-uniform refinement. Resolution was estimated by gold-standard Fourier shell correlation (FSC) between the two independently refined half maps, with a cutoff of 0.143 [51]. Additionally, local resolution was estimated in cryoSPARC using default parameters.

### 4.10. Enzyme-Linked Immunosorbent Assay (ELISA)

Purified proteins were coated on the wells of 96-well microtiter plates at 100 or 200 ng/well and incubated at 37 °C for 2 h. The background was blocked with Blocking solution-3 (WanTai BioPharm, China) at 37 °C for 2 h. Antibodies or sera at 2 mg/mL or 1:100 with two-fold or three-fold serial dilutions were added to the wells (100 µL/well) and incubated at 37 °C for 30 min, followed by five washes. HRP-labeled goat anti-mouse antibody (Abcam, Cambridge, UK) was used as a secondary antibody and incubated at 1:5000 for 30 min. The wells were washed again and catalyzed the reaction using o-phenylenediamine (OPD) as a substrate at 37 °C for 10 min. The optical density at 450 nm was measured on a microplate reader (TECAN, Männedorf, Switzerland).

### 4.11. Immunization in Mice

The six-week-old female BABL/C and C57BL/6 mice were purchased from Shanghai SLAC Laboratory Animal Co., Ltd. All animals were maintained under SPF conditions with controlled illumination, humidity, and temperature, and handled in accordance with the standard use protocols and animal welfare regulations of Xiamen University Laboratory Animal Center.

To estimate the immunogenicity of FR-gE and gE-ST, six-week-old female BABL/C and C57BL/6 mice were randomly divided into groups (n = 5) and immunized intramuscularly at weeks 0, 2 or 0, 2, 4, respectively, with FR-gE and gE-ST diluted in FH002C, XUA01, or aluminum hydroxide adjuvants (50 μL per dose). Blood samples were collected before and at 2-week intervals after vaccination and centrifuged at 13,000 rpm for 10 min to obtain serum samples. Serum samples were preserved at −20 °C before analysis. C57BL/6 mice were sacrificed at the end of the experiment to collect splenocytes for the enzyme-linked immunospot assay (ELISPOT).

### 4.12. VZV Neutralization Assay

The VZV-neutralizing assay was performed as described in a previous study [9]. Serum was inactivated at 56 °C for 30 min and diluted in a two-fold gradient using a protection buffer (9% sucrose, 25 mmol/L histidine, 150 mmol/L NaCl, pH 7.4), and then co-incubated with the virus at 37 °C for 1 h. Subsequently, this mixture was added to ARPE-19 cells that had been pre-seeded in 24- or 48-well plates and incubated at 37 °C with 5% CO_2_ for 1 h. The medium was then replaced with fresh medium. Virus incubated with serum-free protection buffer served as negative controls. After 48 h of culture, an ELISPOT assay was performed, as described in a previous study [52]. The neutralizing titer was determined as the highest serum dilution that could neutralize half of the virus.

### 4.13. ELISPOT

Mouse spleens were collected to prepare the splenocyte single-cell suspensions. A total of 500,000 cells were seeded in each well of a Mouse IFN-γ precoated ELISPOT kit (Dakewe Biotech, Shenzhen, China) and a Mouse IL-2 ELISPOT PLUS kit (Mabtech, Nacka Strand, Sweden) and stimulated with gEgI peptide for 20–24 h. Subsequent steps for IFN-γ and IL-2 analysis were carried out following the instructions provided in the Mouse IFN-γ precoated ELISPOT kit (Dakewe Biotech, China) and Mouse IL-2 ELISPOT PLUS kit (Mabtech, Sweden) manuals, respectively. Spots were counted and analyzed by using CTL-ImmunoSpot S5 (Cellular Technology Limited, Shaker Heights, OH, USA).

### 4.14. Statistical Analysis

GraphPad Prism 9.5.1 was used to analyze the ELISA data and to perform the statistical analysis. Normality of the data distribution was assessed using the Shapiro–Wilk normality test and the Kolmogorov–Smirnov test. A Kruskal–Wallis test with Dunn’s method was applied to analyze differences among more than two groups. *p* values in each group are indicated as * *p* < 0.0332, ** *p* < 0.0021, and *** *p* < 0.0002.

## 5. Conclusions

In our study, we employed the SpyTag/SpyCatcher system to link insect cell-derived gE with ferritin, constructing the nanoparticle antigen FR-gE. We then confirmed that gE could be displayed on the surface of ferritin without affecting particle assembly. Further, we discovered that FR-gE could induce higher neutralizing antibody titers (~2.0-, 2.6-, and 3.2-fold) compared to subunit antigens while formulated with FH002C, aluminum hydroxide, or a liposome-based XUA01 adjuvant. Notably, XUA01-adjuvanted FR-gE induced a significant increase in neutralizing antibody response compared to the live attenuated varicella vaccine and recombinant vaccine, Shingrix. Additionally, we demonstrated that FR-gE could activate effective T-cell responses, producing equivalent levels of IFN-γ and IL-2 cytokines to Shingrix.

## Figures and Tables

**Figure 1 ijms-25-09872-f001:**
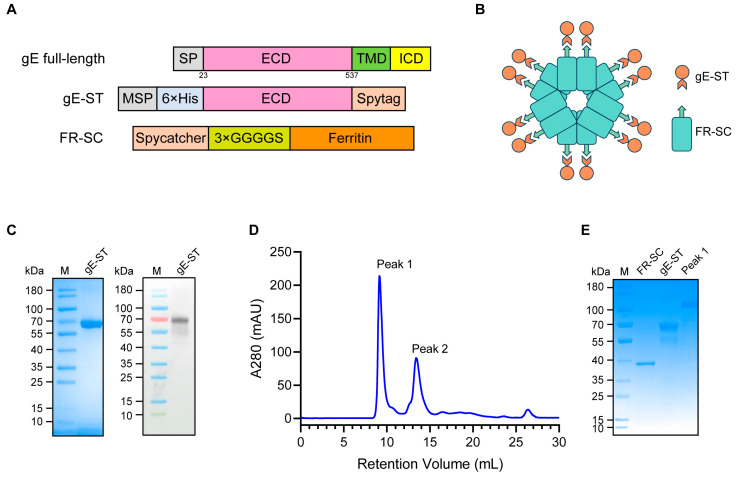
Construction and purification of FR-gE. (**A**) Linear representations of the primary structure of full-length gE and construct designs of gE-ST and FR-SC. ECD (extracellular domain), TMD (transmembrane domain), ICD (intracellular domain), MSP (melittin signal peptide). (**B**) The structural pattern diagram of FR-gE. (**C**) SDS-PAGE and Western blot images of gE-ST. (**D**) Size exclusion chromatography (SEC) purification chromatogram of FR-gE. The main components of peak 1 were collected. (**E**) SDS-PAGE images of proteins purified by SEC. The FR-gE protein was eluted in peak 1.

**Figure 2 ijms-25-09872-f002:**
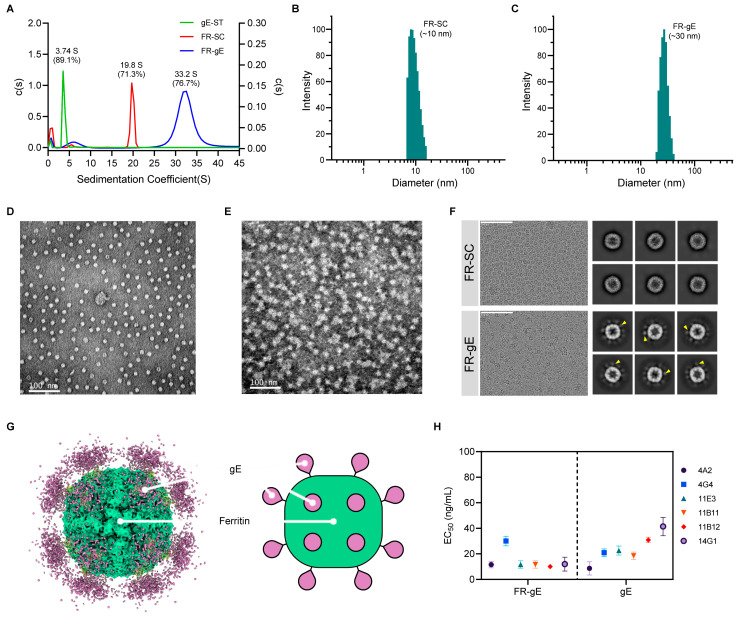
Characterization and structure of FR-gE. (**A**) Analytical ultracentrifugation (AUC) of FR-gE, FR-SC, and gE-ST. Sedimentation velocity analysis was performed using the SedFit method. The curve of FR-gE utilized the right Y axis. (**B**,**C**) Particle size distributions of FR-SC (**B**) and FR-gE (**C**) from dynamic light scattering. (**D**,**E**) Particle morphology of FR-SC (**D**) and FR-gE (**E**) with negative staining under transmission electron microscope (TEM). Yellow arrows indicate gE-ST. Scale bar: 100 nm. (**F**) Cryo-electron microscopy (cryo-EM) raw micrographs (left panels) and two-dimensional (2D) classification (right panels) of FR-SC and FR-gE nanoparticles are shown. Scale bar: 100 nm. (**G**) Cryo-EM density map of the gE–ferritin complex, refined with octahedral symmetry. The FR-SC is depicted in green, and the gE is shown in purple. Schematic corresponding to (**F**), showing the idealized coupling of gE. (**H**) Reactivities of FR-gE and gE with gE mAbs in ELISA. EC_50_ values were calculated by sigmoid trend fitting using GraphPad Prism 9.5.1 software.

**Figure 3 ijms-25-09872-f003:**
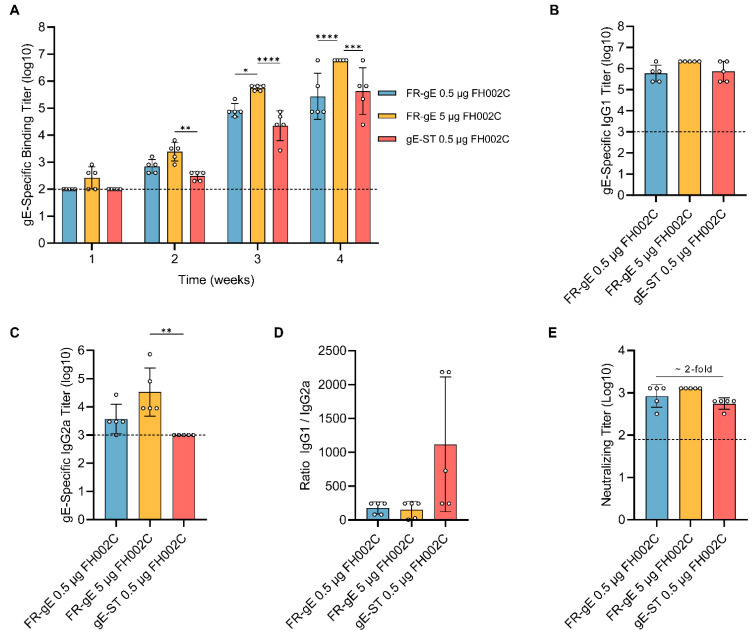
Immunogenicity of FR-gE in BALB/c mice. (**A**) Sera at weeks 1, 2, 3 and 4 were assessed by ELISA for gE-specific binding titers. (**B**,**C**) gE-specific IgG subclass analysis was performed by measuring IgG1 and IgG2a titers using ELISA. (**D**) The ratio of IgG1/IgG2a was calculated for every BABL/c mouse. (**E**) Neutralizing antibody titers of sera on the fourth week were detected by a VZV neutralization assay. The dotted line indicates lower limit of detection (starting serum dilution). All results were analyzed by a Kruskal–Wallis test with multiple comparisons using GraphPad Prism 9.5.1 software; *p* < 0.0332 was considered significant. * *p* < 0.05, ** *p* < 0.01, *** *p* < 0.001, **** *p* < 0.0001.

**Figure 4 ijms-25-09872-f004:**
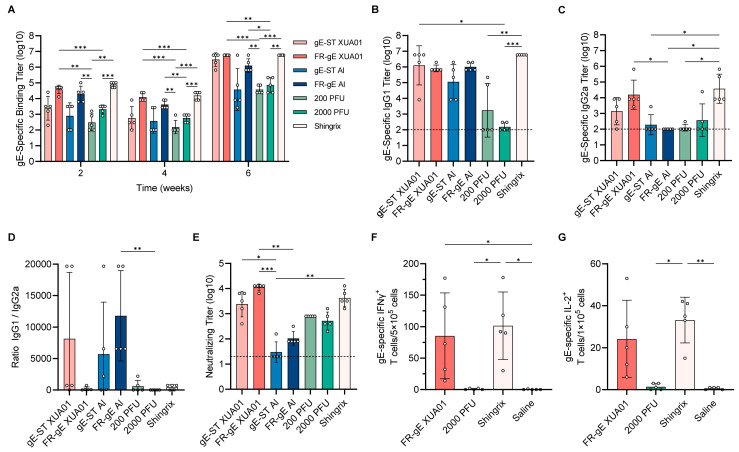
Immunogenicity of FR-gE in C57BL/6 mice. (**A**) Sera at weeks 2, 4, and 6 were assessed by ELISA for gE-specific binding titers. Two gE-ST groups were only compared to two FR-gE groups, and two live attenuated vaccine groups were only compared to two FR-gE groups and Shingrix. (**B**) and (**C**) gE-specific IgG subclass analysis was performed by measuring IgG1 and IgG2a titers using ELISA. (**D**) The ratio of IgG1/IgG2a was calculated for every BABL/c mouse. (**E**) Neutralizing antibody titers of sera on the sixth week were detected by a VZV neutralization assay. (**F**,**G**) The numbers of gE-specific IFN-γ and IL-2 T cells were measured in the groups with lower ratios of IgG1/IgG2a by ELISPOT. The dotted line indicates the lower limit of detection (starting serum dilution). All results were analyzed by a Kruskal–Wallis test with multiple comparisons using GraphPad Prism 9.5.1 software; *p* < 0.05 was considered significant. * *p* < 0.05, ** *p* < 0.01, *** *p* < 0.001.

**Table 1 ijms-25-09872-t001:** Comparison of adjuvanted FR-gE, subunit gE, and live attenuated vaccines.

Vaccine	Animal/Dose	Efficacy
FR-gE/FH002C	BABL/c, SPF /0.5 µg gE/animal	The neutralizing titer of FR-gE is 2 times higher than that of gE-ST.
gE-ST/FH002C		
FR-gE/Al	C57BL/6, SPF /5 µg gE/animal	The neutralizing titer of FR-gE is 2.6 times higher than that of gE-ST. Both FR-gE and gE-ST exhibited lower titers than 200 PFU groups, indicating the low potential of Al-adjuvanted groups as varicella vaccines.
gE-ST/Al		
freeze-dried live attenuated vaccine	C57BL/6, SPF /200 PFU/animal	
FR-gE/XUA01	C57BL/6, SPF /5 µg gE/animal	The neutralizing titer of FR-gE is 3.2 times higher than that of gE-ST and 2.1 times higher than that of Shingrix. The IFNγ and IL-2 cytokine levels of FR-gE are comparable to those of Shingrix.
gE-ST/XUA01		
Shingrix		
freeze-dried live attenuated vaccine	C57BL/6, SPF /2000 PFU/animal	

## Data Availability

The published article includes all datasets generated or analyzed during this study. Any additional information required to reanalyze the data reported in this paper is available from the lead contact upon request.

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
