# Peer review of "Glycoprotein E-Displaying Nanoparticles Induce Robust Neutralizing Antibodies and T-Cell Response against Varicella Zoster Virus"

_ijms, 2024, doi:10.3390/ijms25189872_

Round 1

Reviewer 1 Report

Comments and Suggestions for Authors

1.     In the introduction section, please add a section to provide the characteristic and significance of the glycoprotein 24 E (gE) to explain why the author choose it as the target to induce neutralizing antibodies and T cell response against varicella-zoster viruses (VZV).

2.     In the Material and Method section, please descibe how to prepare the nanoparticles and how to encapsulated the gE in the nanoparticles in details.

3.     For the development of vaccines, safety tests should be prior to efficacy tests. Please show the toxicity test result of FR-gE nanoparticle antigen in mice.

4.     In this article, the authors only provide data and result to prove that FR-gE can induce robust neutralizing antibodies and T cell response in mice. Please provide related procedures and data of challenge tests to confirm that mice immunized with FR-gE are competent to against VZV in the Materials and Method and Result section, respectively.

5.     In the Conclusion section, the authors claimed that “they developed a particulate antigen, FR-gE, based on the VZV glycoprotein E, which can induce stronger immunogenicity in mice compared to subunit antigens and attenuated varicella vaccine, as well as a cellular immune response comparable to that of Shingrix”. Please make a Table to compare the results among FR-gE, subunit antigens and live attenuated vaccine. Additionally, please discuss why FR-gE is better than subunit antigens and live attenuated vaccine in efficacy and safety, especially for safety concerns.

Comments on the Quality of English Language

Moderate editing of English language required.

Author Response

Response to Reviewer Comments on the manuscript [ijms-3158480]:

We thank the reviewer for recognizing the merit of our work and for their suggestions to improve our manuscript. We have fully addressed the comments with appropriate analyses. To facilitate the navigation of this document, we have copied the reviewer’s comments verbatim in blue and typed our responses in black.

Reviewer #1

Comment 1: In the introduction section, please add a section to provide the characteristic and significance of the glycoprotein 24 E (gE) to explain why the author choose it as the target to induce neutralizing antibodies and T cell response against varicella-zoster viruses (VZV).
Response: As suggested, we have added the reasons for selecting gE as a target in the introduction section. The supplemented text is as follows: “The VZV membrane glycoprotein gE plays a crucial role in viral replication and assembly, and it is an indispensable glycoprotein in the process of VZV T cell infection. This also makes it a popular target for VZV vaccine development.” (Page 2, line 69-72)

Comment 2: In the Material and Method section, please describe how to prepare the nanoparticles and how to encapsulated the gE in the nanoparticles in details.

Response: Apologies to the unclear description in the M&M section. In the present Spycatcher/Spytag-based ferritin nanoparticle display system, Spycatcher was engineered into ferritin particles, whereas Spytag was fused with gE protein, therefore gE proteins would be engaged to ferritin particle surface by covalent chemical conjugation instead of encapsulation inside particles. We have articulated the chemical linking in the M&M as follows: “The FR-SC proteins were transferred into ER2566 Competent Cells (Weidi Bio, China) to express the recombinant protein and purified by thermal denaturation at 70 °C for 10 minutes and ammonium sulfate precipitation. The purified gE-ST and FR-SC were co-incubated at 37 °C for 1 h at the molar ratio of 3:1, allowing Spycatcher/Spytag pairing by chemical conjugation, and the mixture were then purified by Superdex 200 increase to isolate FR-gE.” (Page 11, line 356-361)

Comment 3: For the development of vaccines, safety tests should be prior to efficacy tests. Please show the toxicity test result of FR-gE nanoparticle antigen in mice.  
Response: Thank you for pointing out the safety concerns on the nanoparticle system. Fortunately, the safety profile of Ferritin system has been substantially evidenced as excellent in preclinical and clinical trials (NCT03186781), and gE protein has been licensed in the Shingrix vaccine. Thus, we have not yet carried out comprehensive safety evaluation for FR-gE nanoparticle antigens, nevertheless, we found their preferable safety demonstration in all our immunization experiments in mice. We will perform a fully toxicity test package in future once the vaccine candidate enters pre-IND stage and would be generated in GMP condition and reflect more realistic safety data.

Comment 4: In this article, the authors only provide data and result to prove that FR-gE can induce robust neutralizing antibodies and T cell response in mice. Please provide related procedures and data of challenge tests to confirm that mice immunized with FR-gE are competent to against VZV in the Materials and Method and Result section, respectively.

Response: We agree that challenge tests are an important indicator for evaluating vaccine efficacy. However, VZV infects only humans and rare animal model is currently available. Lack of a suitable animal model for VZV infection, the immunogenicity of varicella and zoster vaccines is primarily characterized by neutralizing antibody titers and T cell responses. There are no challenge tests in the latest published articles on the VZV vaccine. We have provided references to recent publications that also lack challenge test data but are recognized in the field:

Jeong, Soo-Kyung et al. “Lipo-pam™ adjuvanted herpes zoster vaccine induces potent gE-specific cellular and humoral immune responses.” NPJ vaccines vol. 9,1 150. 17 Aug. 2024, doi:10.1038/s41541-024-00939-4

Li, Dongdong et al. “Heterologous Prime-Boost Immunization Strategies Using Varicella-Zoster Virus gE mRNA Vaccine and Adjuvanted Protein Subunit Vaccine Triggered Superior Cell Immune Response in Middle-Aged Mice.” International journal of nanomedicine vol. 19 8029-8042. 6 Aug. 2024, doi:10.2147/IJN.S464720

Comment 5: In the Conclusion section, the authors claimed that “they developed a particulate antigen, FR-gE, based on the VZV glycoprotein E, which can induce stronger immunogenicity in mice compared to subunit antigens and attenuated varicella vaccine, as well as a cellular immune response comparable to that of Shingrix”. Please make a Table to compare the results among FR-gE, subunit antigens and live attenuated vaccine. Additionally, please discuss why FR-gE is better than subunit antigens and live attenuated vaccine in efficacy and safety, especially for safety concerns.
Response: Thank you for your suggestion to include a comparative table of the results among FR-gE, subunit antigens, and the live attenuated vaccine, as well as for recommending a discussion on the efficacy and safety of FR-gE. According to your suggestion, we have added the following table in our manuscript. Additionally, we have addressed the safety concerns in comment 3. We appreciate your understanding once again.

Table1. Comparison of adjuvanted FR-gE, subunit gE, and live attenuated vaccines

Vaccine

Animal/Dose

Efficacy

FR-gE/FH002C

BABL/c, SPF

/0.5 µg gE/animal

The neutralizing titer of FR-gE is 2 times higher than gE-ST.

gE-ST/FH002C

FR-gE/Al

C57BL/6, SPF

/5 µg gE/animal

The neutralizing titer of FR-gE is 2.6 times higher than gE-ST. Both FR-gE and gE-ST exhibited lower titers than 200 PFU groups, indicating the low potential of Al-adjuvanted groups as varicella vaccines.

gE-ST/Al

freeze-dried live-attenuated vaccine

C57BL/6, SPF

/200 PFU/animal

FR-gE/XUA01

C57BL/6, SPF

/5 µg gE/animal

The neutralizing titer of FR-gE is 3.2 times higher than gE-ST and 2.1 times higher than Shingrix. The IFNγ and IL-2 cytokine levels of FR-gE is comparable to those of Shingrix.

gE-ST/XUA01

Shingrix

freeze-dried live-attenuated vaccine

C57BL/6, SPF

/2000 PFU/animal

Reviewer 2 Report

Comments and Suggestions for Authors

The manuscript entitled “Glycoprotein E-displaying nanoparticles induce robust neutralizing antibodies and T cell response against varicella-zoster virus” by Wang et al. described the construction of Glycoprotein E-displaying nanoparticles FR-gE and successfully induced substantial gE-specific binding and VZV neutralizing antibody responses in BALB/c and C57BL/6 mice. Co-treatment of FR-gE with the liposome-based XUA01 adjuvant (FR-gE/XUA01) induced further increase in neutralizing antibody response and generation of IFN-γ and IL-2 levels comparable to those induced by Shingrix. The following are my comments and suggestions:

1.     In Figure 3, FH002C was used as an adjuvant for the FR-gE and gE-ST in the immunogenicity assay of BALB/c mice, while in Figure 4, aluminum hydroxide and XAU01 were used as adjuvants for the FR-gE and gE-ST in the immunogenicity assay of C57BL/6 mice. Please explain why different adjuvants and different mice species were used.

2.     In Figure 4, inclusion of FH002C as an adjuvant for the immunogenicity assay of C57BL/6 mice (Figure 4) are suggested.

3.     On page 10, line 352 to line 355, the authors described that “Throughout the experiment, both gE-ST and FR-gE paired with XUA01 adjuvant elicited stronger immune responses than those paired with aluminum hydroxide adjuvant, with average titers in the 2000 PFU and 200 PFU groups remaining at a lower level of 4-5 log at week 6.” (Figure 4A). To know whether these differences display statistics significant, statistical analyses between gE-ST/XUA01 and gE-ST/AI, and between FR-gE/XUA01 and FR-gE/AI should be performed.

4. On page 11, line 380 to line 383, the authors described that “Mice immunized with FR-gE/XUA01 displayed the highest neutralizing antibody titers, significantly surpassing the gE-ST/Al and FR-gE/Al groups, with increases of approximately 3.2-fold compared to the gE-ST/XUA01 group and 2.1-fold relative to Shingrix.” (Figure 4E). To know whether these differences display statistics significant, statistical analyses between FR-gE/XUA01 and gE-ST/XUA01, and between FR-gE/XUA01 and Shingrix should be performed. Similar phenomena also occur in the other places.

Author Response

Response to Reviewer Comments on the manuscript [ijms-3158480]:

We thank the two reviewers for recognizing the merit of our work and for their suggestions to improve our manuscript. We have fully addressed the comments with appropriate analyses. To facilitate the navigation of this document, we have copied the reviewers’ comments verbatim in blue and typed our responses in black.

Reviewer #2

Comment 1: In Figure 3, FH002C was used as an adjuvant for the FR-gE and gE-ST in the immunogenicity assay of BALB/c mice, while in Figure 4, aluminum hydroxide and XAU01 were used as adjuvants for the FR-gE and gE-ST in the immunogenicity assay of C57BL/6 mice. Please explain why different adjuvants and different mice species were used.

Response: FH002C is a innovative adjuvant developed by our laboratory, which has not yet been applied in licensed vaccines. Another study from our laboratory showed that FH002C has a notable advantage in inducing neutralizing antibodies in BALB/c mice. Therefore, we conducted comparisons using FH002C in BALB/c mice. To further evaluate the effect of FR-gE on stimulating cellular immunity, we chose C57 mice, which are more sensitive to cell-mediated immunity, and the XUA01 adjuvant, which exhibited more evident effects, for immune evaluation. To clarify this point more effectively, we have revised the original text: “To further evaluate particulate gE as an antigen for both varicella and shingles vaccines, we designed a combination of two well-characterized adjuvants: the traditional aluminum adjuvant for varicella and the potent XUA01 adjuvant (a mimic of licensed liposome-based AS01B adjuvant) for shingles, and compared it with the varicella attenuated vaccine, shingles attenuated vaccine, and recombinant shingles vaccine. For further detection of cellular immune responses, we performed immunization assay in C57BL/6 mice (n=5 per group), administering gE-ST and FR-gE in combined with XUA01 adjuvant as well as aluminum hydroxide adjuvant at weeks 0 and 4.” (Page 7, Line 217-224)

Comment 2: In Figure 4, inclusion of FH002C as an adjuvant for the immunogenicity assay of C57BL/6 mice (Figure 4) are suggested.
Response: Thank you for the suggestion. Considering the relative lower effectiveness of FH002C for T cell response in our previous study (Wu et al., STM, 2021), we therefore didn't include the FH002C as a potential adjuvant, but only used AS01b-like XUA01 adjuvant in the experiments of Figure 4.

Comment 3: On page 10, line 352 to line 355, the authors described that “Throughout the experiment, both gE-ST and FR-gE paired with XUA01 adjuvant elicited stronger immune responses than those paired with aluminum hydroxide adjuvant, with average titers in the 2000 PFU and 200 PFU groups remaining at a lower level of 4-5 log at week 6.” (Figure 4A). To know whether these differences display statistics significant, statistical analyses between gE-ST/XUA01 and gE-ST/AI, and between FR-gE/XUA01 and FR-gE/AI should be performed.
Response: As suggested, we have added the description of the statistical analyses. The revised text is as follows: “Throughout the experiment, both gE-ST and FR-gE paired with XUA01 adjuvant elicited stronger immune responses than those paired with aluminum hydroxide adjuvant, with no significant difference. At week 6, the average titers in the 2000 PFU and 200 PFU groups remaining at a lower level of 4-5 log, significantly lower than those of FR-gE with aluminum hydroxide adjuvant or XUA01.” (Page 7, Line 233-237)

Comment 4: On page 11, line 380 to line 383, the authors described that “Mice immunized with FR-gE/XUA01 displayed the highest neutralizing antibody titers, significantly surpassing the gE-ST/Al and FR-gE/Al groups, with increases of approximately 3.2-fold compared to the gE-ST/XUA01 group and 2.1-fold relative to Shingrix.” (Figure 4E). To know whether these differences display statistics significant, statistical analyses between FR-gE/XUA01 and gE-ST/XUA01, and between FR-gE/XUA01 and Shingrix should be performed. Similar phenomena also occur in the other places.
Response: As suggested, we have added the description of the statistical analyses. The revised text is as follows: “Mice immunized with FR-gE/XUA01 displayed the highest neutralizing antibody titers, significantly surpassing the gE-ST/Al and FR-gE/Al groups, with no significant increases of approximately 3.2-fold compared to the gE-ST/XUA01 group and 2.1-fold relative to Shingrix.” (Page 8, Line 263-266)

Round 2

Reviewer 1 Report

Comments and Suggestions for Authors

The manuscript has been significantly improved.

Comments on the Quality of English Language

No.

Reviewer 2 Report

Comments and Suggestions for Authors

The authors have fully addressed my concerns.